# A Mini Review of *S*-Nitrosoglutathione Loaded Nano/Micro-Formulation Strategies

**DOI:** 10.3390/nano13020224

**Published:** 2023-01-04

**Authors:** Hui Ming, Kunpeng Zhang, Shengbo Ge, Yang Shi, Chunan Du, Xuqiang Guo, Libo Zhang

**Affiliations:** 1State Key Laboratory of Heavy Oil Processing, College of Engineering, China University of Petroleum-Beijing at Karamay, Karamay 834000, China; 2Jiangsu Co-Innovation Center of Efficient Processing and Utilization of Forest Resources, International Innovation Center for Forest Chemicals and Materials, College of Materials Science and Engineering, Nanjing Forestry University, Nanjing 210037, China; 3Faculty of Chemical Engineering, Shandong Institute of Petroleum and Chemical Technology, Dongying 257000, China

**Keywords:** *S*-nitrosoglutathione, formulation, encapsulation, strategies

## Abstract

As a potential therapeutic agent, the clinical application of *S*-nitrosoglutathione (GSNO) is limited because of its instability. Therefore, different formulations have been developed to protect GSNO from degradation, delivery and the release of GSNO at a physiological concentration in the active position. Due to the high water-solubility and small molecular-size of GSNO, the biggest challenges in the encapsulation step are low encapsulation efficiency and burst release. This review summarizes the different nano/micro-formulation strategies of a GSNO related delivery system to provide references for subsequent researchers interested in GSNO encapsulation.

## 1. Introduction

*S*-nitrosoglutathione (GSNO) is one of the most biologically abundant nitric oxide (NO) donors [1]. GSNO function as a bioactive molecule mainly through the delivery NO. NO is a messenger molecule that plays a vital role in many physiological processes, including cardiovascular system regulations, neurotransmissions and immune system functions [2,3,4]. It has anti-inflammatory properties and can also act as an antioxidant to protect cells from damage caused by free radicals [5,6]. Additionally, nitric oxide helps to regulate cell growth and cell death which makes it useful for cancer therapy [7,8]. GSNO is known as a physiological form of NO storage and a component of NO-dependent signal transduction pathways [1]. Due to the apparent drawbacks of an extremely short half-life and rapid diffusion without focused target behavior in NO, its use in therapy is limited [9]. In order to solve these problems, different NO donors have been developed. Compared to traditional NO donors, including organic nitrates, N-diazenium-diolates, nitro-benzenes, furoxa and metal nitrosyl complexes [10,11,12,13], GSNO intrinsically exists in an organism, is a better candidate, with no tolerance phenomenon observed and low toxicity [14,15]. As a NO donor, GSNO exhibits better stability and bioactivity. However, its clinical application is still limited because of the high reactivity of the S-NO bond [16]. Besides, concentrations and the targets to which GSNO is delivered influence efficiency [17]. Therefore, in order to protect GSNO from environmental decomposition and to release NO in the expected concentrations and location, it is crucial to develop different formulations to deliver GSNO for further pharmaceutical applications. Nano/micro-materials can help deliver NO to the target site in a controlled and prolonged manner, reducing the side effects on normal tissues and increasing therapeutic efficiency, representing a very good choice for the application of GSNO (an endogenous molecular), which is acknowledged as one of the best potential NO donors for its clinical efficiency. Since instability and challenges when encapsulating GSNO exist, more effort should be made into developing further clinical applications of GSNO. Until now, there is no literature reporting an overall picture of GSNO delivery systems. Thus, this review provides a summary of GSNO related nano/micro-formulation strategies. The comparison between different strategies of different GSNO delivery systems is made through an overview of the current literature. Researchers can easily choose the most appropriate strategy and method according to their requirements after reading this review. Its objective is to inspire related researchers to further clinical exploration of GSNO applications.

After careful research and a literature analysis, the strategies of GSNO delivery systems can be classified as shown in Figure 1.

(1) Direct GSNO loaded nano/micro-formulation: GSNO is encapsulated directly into the formulation.

(2) *S*-nitrozation of GSH loaded nano/micro-formulation: Glutathione (GSH, a precursor of GSNO) is encapsulated into the formulation, followed by *S*-nitrozation of the thiol group, presented in or on the formulation.

(3) GSNO conjugated nano/micro-formulation: GSNO is firstlly conjugated with polymer, followed by encapsulation into different formulations.

## 2. Direct GSNO Loaded Nano/Micro-Formulation

The direct encapsulation strategy is the simplest and most common method in drug delivery systems. In this strategy, GSNO, as a potential therapeutic agent, is directly incorporated into different formulations, such as liposomes, nanoparticles/microparticles and nanocomposites.

### 2.1. Liposomes

Liposomes are small closed vesicles with a bilayer-wrapped aqueous phase structure, formed by dispersing phospholipids or other lipids in water, which can also be called an artificial biofilm because its structure is similar to that of a biological membrane. As lipid-based drug carriers, liposomes are widely valued and studied, not only because the main raw material for their preparation, phospholipids, is an intrinsic component of human cells, which makes them biocompatible without immunogenicity, but also because they can be prepared as nanoscale particles. This makes them easier to pass through biological barriers such as blood vessel walls and cell membranes [18]. Due to the specific core-shell structure formed by the lipid bilayer, both hydrophilic (core) and hydrophobic (shell) drug candidates can be encapsulated in liposomes to achieve reduction in toxicity and prolonged duration of the therapeutic effect.

In order to potentiate GSNO antibacterial activities, GSNO achieved high local and sustained intracellular concentration in macrophages by encapsulation in liposomes, which leads to drug delivery systems for targeting macrophages because of their natural recognition by macrophages [19]. Owing to GSNO stability issues, the solvent-spherule evaporation method, a simple and scalable method without heating or freezing steps, is applied to maintain GSNO integrity. The final sterically stabilized cationic liposomes (SSCL), which are prepared by hydrogenated soybean phosphatidylcholine, cholesterol, stearyl-amine, N-(carbonyl-methoxy-polyethylene glycol-5000)-1,2-dipalmitoyl-sn-glycero-3-phosphoethanolamine sodium salt, obtain narrowly distributed and stable nanosized particles (size 156 ± 10 nm, zeta-potential 16 ± 1 mV) displaying encapsulation efficiency up to 24 ± 2% with in vitro sustained release (no more than 28 ± 7% over 8 h). Compared to free GSNO, GSNO-loaded SSCL shows significantly higher intracellular NO uptake, by a factor of 20. Anti-bacterial experiments indicate that GSNO-loaded SSCL displays strong bacteriostatic effects on Staphylococcus aureus and Pseudomonas aeruginosa, which make the delivery system possible in the treatment of pulmonary infectious diseases. However, the stability of the liposomes is not addressed. Additionally, it is possible that there are issues with toxicity when used on humans or animals, which will need to be further investigated before therapeutic use.

To simultaneously overcome tumor vascular endothelium and extracellular matrix barriers, GSNO is co-encapsulated with anti-cancer drug doxorubicin (DOX) loading palyamido-amine small particles-PAMAM@DOX (10 nm) in ultrasound responsive liposomes that possess an air layer between the lipid bilayer (as shown in Figure 2) [20]. The ultrasound liposomes are prepared by thin film evaporation combined with frozen-thaw method with components of dipalmitoyl phosphatidylcholine, dioleoyl phosphatidylethanolamine, 1,2-dipalmitoyl-sn-glycero-3-phospho-(1′-rac-glycerol) and cholesterol, leading to particle size of around 422 nm and zeta-potential of about −61.2 mV. However, the encapsulation efficiency is not indicated in this study. Under ultrasound stimulation, GSNO rapidly produces NO to act on tumor vascular smooth muscle, leading to tumor vasodilation, while the rupture of the gas layer in the liposome disrupts the lipid bilayer, resulting in the dramatic release of encapsulated PAMAM@DOX within the tumor vasculature and subsequent extravasation through the endothelial cell gap to penetrate deep into the tumor [20]. Meanwhile, the negatively charged tumor cell membrane tends to bind to the cationic PAMAM@DOX, promoting cellular internalization. Eventually, compared with only DOX-loaded conventional liposomes, GSNO- PAMAM@DOX co-loaded ultrasound responsive liposomes augment the tumor therapeutic efficiency by 32.5% and 56.5% in the treatment of MCF-7 and MiaPaCa-2 tumors [20].

### 2.2. Polymeric Nano/Micro-Particles

Eudragit^®^ nano/micro-particles are reported to encapsulate GSNO adapted to oral administration for cardiovascular disease with sustained release, intracellular delivery or enhanced permeability trough the intestinal barrier [21,22,23,24].

Owing to the proved intestinal barrier permeability [25], GSNO is a good candidate for oral administration to treat diseases related to NO deficiency (e.g., atherosclerosis, pulmonary hypertension, thrombosis, ischemia [26,27,28,29]). Zhou et al. reported three types of Eudragit^®^ polymer particles with a similar encapsulation efficiency (around 30%): nanoparticles (NPs) (0.225 ± 0.003 μm) and microparticles (MPs) (69 ± 7 μm) prepared by a water-in-oil-in-water (W/O/W) double emulsion-evaporation process versus microparticles (165 ± 14 μm) prepared by a solid-in-oil-in-water (S/O/W) emulsion/evaporation process [24]. Slightly slower in vitro release is observed with MPs than NPs. The in vitro intestine permeability study shows that GSNO loaded MPs displayed two times more of NO permeability compared to GSNO loaded NPs. This increased permeability is considered to be due to an enhanced absorption rate caused by the particle size and surface characteristics, which allow for better interaction between the particles and cell membranes in order for them to pass through more easily. This study only provides data on the preparation and storage of particles encapsulating GSNO, not its effectiveness as a drug delivery system. Further preclinical trials are needed to determine if these particles can effectively deliver GSNO via an oral administration.

To deliver appropriate NO concentrations to target sites with long-lasting effect, Eudragit^®^ RL NPs (289 ± 7 nm) are designed to encapsulate GSNO with a final efficiency of 54% [21]. The NPs are fabricated by a W/O/W double emulsion-evaporation process. Sustained release is proved by delayed maximum protein *S*-nitrozation (a marker of NO bioavailability) until 18 h for GSNO-NPs in contact with smooth muscle cells, compared with 1 h for control group (free GSNO). Interestingly, GSNO-NPs release 100% of encapsulated GSNO within only 3 h in an in vitro release study. The authors hypothesize that the presence of proteins in the fetal bovine serum (mostly albumin) surrounding the NPs which leads to the formation of a ‘‘biological wall’’ may slow down the diffusion rate of GSNO throughout the NPs [21]. Another study shows that Eudragit^®^ RL NPs entered human monocyte cells by clathrin- and caveolae-mediated endocytosis, which facilitates intracellular delivery of GSNO to stimulate immune defenses [30]. The limitation of the reported GSNO-loaded nanoparticles is that they can only provide protection and sustained release of NO for up to 18 h when exposed to smooth muscle cells. This may not be long enough for some treatments.

Eudragit^®^ microparticles adapted to oral administration of GSNO in treatment of inflammatory bowel disease are designed with a pH sensitive ability to achieve controlled release in the intestine instead of in an acid environment in the stomach [31]. GSNO is reported to prevent intestinal barriers from disruptions in in vitro, ex vivo and in vivo study [32,33,34,35], which makes it an alternative drug for treatment of inflammatory bowel disease. For a proposal of oral administration of GSNO to treat inflammatory bowel diseases, GSNO is encapsulated (82% ±1%) in Eudragit^®^ FS 30D (pH sensitive polymer) MPs (5 ± 1 μm) by spray-drying method with optimized parameters (inlet temperature 120 °C, outlet temperature 47 °C, solvent flow 5 mL/min and air flow 100%) to protect GSNO from degradation in the stomach (acidic pH and presence of digestive enzymes), but finally realizes significant release in the intestine (action site, pH 6.8–7.4) [31]. However, the prepared microparticles may not provide sustained release, as it is observed that most of the drug is released within a few hours.

Poly (lactic-co-glycolic acid) (PLGA) microparticles are reported to encapsulate GSNO with a revealed sustained release profile for their wound healing and antibacterial application or subcutaneous administration because of vasorelaxant effect [36,37,38].

In order to achieve prolonged release of NO via application in order to accelerate the healing of Methicillin-resistant Staphylococcus aureus (MRSA)-infected wounds, PLGA microparticles (158.7 ± 20.5 μm) are prepared by a S/O/W emulsion-solvent evaporation method (Figure 3) to encapsulate GSNO with a final encapsulation efficiency of 57.6% ± 6.5% [36]. However, the method used to create the microparticles may be difficult to scale up for large-scale production. The in vitro release experiment shows more than 7 days sustained release of NO. Remarkable antibacterial efficacy *in cellulo* and acceleration of MRSA-infected wound healing in vivo are found by treatment with GSNO-MPs.

In situ, PLGA MPs [37,38] are reported to deliver GSNO under subcutaneous administration in a sustained manner for the treatment of stroke because of GSNO’s vasodilation effect [39], along with a potent inhibitory effect of platelet aggregations [40]. In order to reduce the initial burst release of GSNO, PLGA dissolved in the organic solvent (N-methyl-2-pyrrolidone (NMP) or triacetin (TA)) is used as GSNO dispersed substrate. An additional emulsification process with an external phase (sesame oil 96%, aluminium monostearate 2% and Span 80 2% m/m) is carried out to reduce viscosity in order to facilitate injection [37]. GSNO loaded in situ MPs are obtained by subsequent injection of the emulsion into an aqueous phase resulting in solidification of the emulsion droplets by solvent/water exchange. A linear release of GSNO from in situ MPs loaded with 5% GSNO m/m is observed lasting for 24 h in in vitro release study [38]. Sustained release is achieved through drug diffusion and polymer degradation [37]. The in vivo study further proves sustained delivery of GSNO in in situ MPs by longer (at least 2 days) reduced effects on arterial pressure compared with control group (free GSNO) [38]. The main limitation of the in situ MPs encapsulating GSNO is that the size of the particles are not controllable and uniform. Additionally, there may be other factors to consider when using this treatment such as potential side effects or interactions with other medications.

### 2.3. Inorganic Nanoparticles

CaCO_3_−mineralized nanoparticles (MNPs) are reported to co−delivery GSNO and doxorubicin (DOX) for cancer treatment along with increased GSNO stability and anticancer activity of DOX [41]. GSNO is successfully encapsulated into CaCO_3_−MNPs to form pH-sensitive nanocarriers that dissolve at acidic endosomes to trigger intracellular release of nitric oxide (NO). The GSNO−MNPs (248.8 ± 12.1 nm, loading amount 13.3%wt) are carried out by a PEG−PAsp−templated in situ mineralization method (Figure 4). During the mineralization process, ionic GSNO can be attached in situ inside the CaCO_3_ core, which greatly increases the stability of GSNO. The NO is produced in the cytoplasm from GSNO which is released from MNPs by dissolving at endosomal pH. NO generated by the GSNO−MNPs augments the therapeutic efficiency of DOX in in vitro cell experiments. This could can be explained by NO’s ability to inhibit tumor growth by promoting apoptosis [42]. The main shortcoming is that GSNO−MNPs are not able to release NO quickly enough at physiological pH. This is due to the fact that their CaCO_3_ core does not dissolve easily at physiological pH, which limits their ability to release NO. Besides, their efficacy and safety in vivo has yet to be determined.

### 2.4. Nanocomposites

Nanocomposites are composites consisting of nanoscale particles dispersed in the polymer matrix [43]. In order to reinforce the stability or further prolong the release profile of GSNO or a specific delivery target, different drug delivery systems or strategies were mixed to encapsulate GSNO, such as ZIF-8@Cytomembrane nanoplatform [44], metal organic framework (HKUST-1@MIL-100(Fe)) nanocomposites [45], chitosan nanoparticles incorporated in Pluronic F127 hydrogel, Ag nanoparticles in PVA/PEG film [46], Eudgragit@RL nanoparticles in alginate/chitosan microparticles [22,23], and so on.

Zeolitic imidazolate framework 8 (ZIF-8), which is self-assembled from Zn^2+^ and 2-methylimidazolate, has attracted more and more attention in drug delivery fields because of its various merits such as excellent thermal and chemical stability, high loading capacity, high specific surface area, adjustable pore size and pH-sensitive degradability [47].

To enhance the sonodynamic therapy effect on the tumor, GSNO (EE 53.1%) and chlorin e6 (Ce6, EE 72.8%) are co-delivered by embedding in the dual pH/ultrasound response biomimetic ZIF-8@Cytomembrane nanoplatform (212 nm), which is prepared by coating ZIF-8 with homologous tumor cell membrane for tumor-targeting purposes (Figure 5) [44]. This so-called gas-sonodynamic combined therapy represents a more effective cancer treatment model through accumulating probes at the targeted tumor site and releasing Ce6 and GSNO by degradation of ZIF-8 in the acidic microenvironment of the tumor. Followed by ultrasound stimulation, NO produced from GSNO and reactive oxygen species generated by Ce6 react with each other to produce highly reactive peroxy-nitrite molecules and other reactive nitrogen species to achieve a better ability to kill tumor cells [44].

For the purpose of developing an ideal technology for cancer treatment (reducing side effects and promoting therapeutic efficiency), the rational design of a smart stimuli-responsive and fully biodegradable delivery system of core-shell MOF (HKUST-1@MIL-100(Fe)) nanocomposites by using ZIF-8 nanoparticles as a sacrificial template and co-encapsulating black phosphorous quantum dots (BPQDs) and GSNO is reported [45]. As shown in Figure 6, BPQDs are firstly loaded into ZIF-8 NPs that exhibits excellent adsorption toward Cu^2+^ and completely degrades in an acidic environment, which acts as a sacrificial template to avoid BPQD degradation in the presence of Cu^2+^. After Cu^2+^ is anchored within the pores and cage of ZIF-8 to fix the reaction region, the organic linker-H_3_BTC is added to stimulate the self-assembly of HKUST-1 NPs around the localized region of Cu^2+^ and the coinstantaneous degradation of ZIF-8, which is called a “Framework exchange” strategy. The MIL-100(Fe) shell is wrapped around the BPQDs loaded HKUST-1 core by an epitaxial growth method which Fe^3+^ adsorbs on the HKUST-1 NPs, then coordinating with the organic linker-H_3_BTC to prepare core-shell MOF (HKUST-1@MIL-100(Fe)) nanocomposites. The mesoporous MIL-100 shell further encapsulates GSNO to form the final nano-formulation. The weak environment, such as the presence of lysosomes and peroxisomes, triggers the release of Fe^3+^ and Cu^2+^ after the internalization of nanocomposites. The released Fe^3+^ and Cu^2+^ react with GSH to produce Fe^2+^ and Cu^+^, which generates ·OH in the presence of H_2_O_2_. Meanwhile, Cu^+^ stimulates NO production by decomposing GSNO. These NO and ·OH then attack the mitochondria and DNA in cancerous cells with less side effects than conventional chemotherapy drugs, because of their limited half-life [42,48,49,50,51]. Furthermore, the generation rate of NO and ·OH can be speeded up by increased temperature caused by BP under near-infrared (NIR) irradiation. Thus, the GSNO-BP co-loaded MOF nanocomposites design represents an ideal treatment for cancer with a fully biodegradable metabolic pathway and multimodal therapy. This formulation uses NIR irradiation and the microenvironment of tumor tissues to create an active and passive targeting response to achieve an accurate therapy with high therapeutic efficiency and low side effects.

Lee et al. prepared GSNO delivery and catalysis combination poly- (lactic-co-glycolic acid) (PLGA) polymersome by double emulsion(W/O/W) method encapsulating GSNO in the inner water phase core and incorporating gold nanoparticles (Au NPs) in the oil phase membrane, respectively [46]. NO generation is realized by reaction between GSNO and Au NPs under photothermal heating caused by breakdown of polymersomes’ condition, leading to selective cerebral vasodilation in zebrafish head [46]. This formulation can be delivered with clinical intravascular optical catheters to enable light stimulated local NO generation, which avoids side effects associated with systemic administration.

GSNO is designed to be encapsulated into the polymeric nanocomposite platform for dermatological application with a sustained release profile, because of its effects on dermal blood flow, skin defense and tissue repair. In order to prepare a formulation that will release NO directly at the target site, GSNO was firstly encapsulated into chitosan nanoparticles (CS NPs) (112.2 ± 2.22 nm), followed by further entrapment in Pluronic F-127 (PL) hydrogels (Figure 7) [52]. The result indicates that the combination of GSNO-CS NPs and a PL hydrogel may be useful for dermatological applications due to their mechanical properties and ability to release therapeutic amounts of NO at the desired application site. The CS NPs act as drug delivery system with mucoadhesive properties, while the PL hydrogel allows for prolonged contact time of the NO donor on skin or mucosal surfaces.

For direct application of wound dressing on solid tumors, GSNO is entrapped together with AgNPs in the polyvinyl alcohol (PVA)/polyethylene glycol (PEG) film by a solvent casting method [53]. Sustained release of GSNO and AgNPs is observed through the in vitro diffusion profile. Compared with GSNO or AgNPs’ individual encapsulation formulation, the PVA/PEG film, co-encapsulating GSNO and AgNPs, shows significantly increased cytotoxic effect against human cervical carcinoma and human prostate cancer cell lines. Besides, considering the proven antibacterial activity, GSNO/AgNPs PVA/PEG film can be applied to treat infected wounds on solid tumor or used as antibacterial coating on medical devices.

To summarize the direct GSNO encapsulation strategy, it is easily concluded that, compared with free GSNO, the GSNO nano/micro-formulations significantly enhance therapeutic efficiency or augment co-delivered drug efficiency by sustained/controlled release or targeting strategy. The encapsulation strategy also makes its therapeutic function possible by increasing its stability. Table 1 shows the lists of direct GSNO loaded nano/micro-formulations without nanocomposites, because nanocomposites are simply double encapsulations by mixing single encapsulation methods listed in the table to further increase the efficiency of the delivery system. Liposomes are a good choice for sustained GSNO release and enhanced NO cellular uptake to increase this final therapeutic efficiency but their GSNO encapsulation efficiency is quite low compared with the other formulation types. This may be explained by the high solubility and diffusion rate of GSNO in water and its complicated and long duration preparation method, which leads to GSNO leakage and decomposition. The CaCO_3_ nanoparticles present a considerable loading amount of GSNO and increase the stability of GSNO, but the application of this formulation is very limited because it can only release GSNO in a specific environment. Polymer nano/micro-particles represent the most frequent choice, because they exhibit acceptable encapsulation efficiency and expected functions such as sustained/controlled release or increased intracellular delivery of NO. Among all these polymer nano/micro-particles, the double emulsion-evaporation process is always used because of its resulting relatively high encapsulation efficiency in encapsulating hydrophilic small molecules. Additionally, in order to obtain longer release of GSNO, the polymer choice is a key factor. Comparing the release profile of the nano/micro-particles prepared by Eudragit@RL or RS and PLGA, which are considered as sustained release polymer material, PLGA nano/micro-particles present as a better candidate for longer release requirements [22,24,30,31,36,37,38]. However, due to the complicated preparation step of the double emulsion-evaporation process, it is not fit for scaling up in comparation with the spray-drying method. Overall, in order to prepare a longer GSNO release formulation fitting for industry production, using PLGA as material by spray-drying method can be a good choice for direct GSNO encapsulation.

## 3. *S*-Nitrozation of GSH Loaded Nano/Micro-Formulation

Due to the instability of the S-NO presented in GSNO, it may be decomposed during the GSNO formulation’s preparation and manipulation, such as by light and heat [54,55]. In order to decrease the possibility of GSNO chemical decomposition as much as possible, another option for a GSNO encapsulation strategy is entrapping the precursor of GSNO (GSH) in the dosage form and further reaction by *S*-nitrozation of the free thiol group of encapsulated GSH to form GSNO.

A novel method of using mucoadhesive polymers (alginate/chitosan) to encapsulate GSH, which can then be *S*-nitrozated to obtain GSNO, results in GSNO encapsulated in biodegradable and non-toxic NPs with sustained release of NO [56,57,58]. The NPs are prepared by polyelectrolyte complexation that refers to a simple mixing process of different charged polymers solution to form a stable complex. The rate of NO release from GSNO inside these nanoparticles is much slower than if it were released in its free form. The particle size and zeta-potential are influenced by the ratio of alginate and chitosan. Stability study and cytotoxicity study show that the positive nanoparticles have a better stability and a lower toxicity than the negative ones. Additionally, it is found that coating NPs with polymers or using stabilizing agents such as surfactants can reduce their toxicity levels even further.

Chitosan nanoparticles (CS NPs) are reported to increase GSNO stability and absorption of NO through skin [59,60]. GSH loaded chitosan NPs are prepared by ionotropic gelation process, followed by nitrozation of GSH-CS NPs leading to the formation of GSNO-CS NPs. The combined treatment of GSNO-CS NPs with UV irradiation leads to the increase of NO and *S*-nitroso-thiol levels in human skin.

SPIONs possess well-defined morphological, structural and magnetic characteristics, in addition to high stability and biocompatibility in biological environments, leading to them becoming potential candidates for drug delivery. It has been reported that the surfaces of the superparamagnetic iron oxide nanoparticles (SPIONs) are functionalized with tripeptide GSH or poly-(ethylene glycol) (PEG), which improves their dispersion and biocompatibility in aqueous/biological environments [61,62]. Furthermore, free thiol groups on the surface of GSH-SPIONs can be nitrozated, leading to *S*-nitrozated SPIONS that act as a nitric oxide donor, which makes them useful for therapeutic purposes such as inhibiting platelet adhesion, accelerating wound healing and increasing blood flow [63,64,65]. It is observed that GSNO SPIONS released up to 124 μmol of NO/g, while PEG -GSNO-SPIONS released 33.2 μmol of NO/g. Therefore, GSH-SPIONs/PEG-GSH-SPIONs may serve as targeted nano-vehicles or diagnostic tools in biomedical applications.

The literature on encapsulation of GSH firstly and then *S*-nitrozation after the formulation step reports a useful concept for avoiding GSNO decomposition during the formulations preparation step. The problem is that all the literature demonstrates the efficient encapsulation efficiency of GSH, but have not indicated the yield of *S*-nitrozation. After all, the *S*-nitrozation reaction yield and rate are highly dependent on the nitrozation reagent diffusion amount and rate inside the formulation, which represent a natural barrier. In addition, if the nitrozation reagent is easy to diffuse through the formulation wall, GSH can also diffuse easily outside the formulation during the S-nitrozation step. Besides, if we want to discover the exact yield of S-nitrozation, we will waste a lot of time due to GSNO in the formulation. In our opinion, it is not an appropriate strategy for GSNO encapsulation efficiency.

## 4. GSNO Conjugated Nano/Micro-Formulation

Since GSNO diffuses easily because of its small hydrophilic molecule property, the previous encapsulation strategies for GSNO loaded nano/micro-formulations have the risk of leakage of GSNO or GSH during preparation steps and limited sustained release of GSNO. Considering these problems, to create a novel, stable and biodegradable NO donor polymer by conjugation of GSNO with polymer could be a new choice.

It is reported that the increased stability and sustained release of GSNO was proven after conjugation to chitosan (CS) [66]. The chitosan backbone provides a more stable environment for NO delivery than free GSNO. Additionally, this polymer has been designed to be biodegradable and non-toxic in order to ensure safe use in human bodies. The loading capacity of this novel NO donor polymer reached up to 525.08 ± 151.35 μmol of NO /g polymer. The ex vivo study using a chamber exhibits the sustained release ability of GSNO-CS and reserve ability in the mucosal side of the intestine, which makes it a good choice for oral delivery in Crohn’s disease treatment. For the same application, alginate is also used to conjugate GSNO, which leads to a similar loading capacity of 468 ± 23 μmol/g polymer with observed extended-release effect and long-term stability [67].

Even though the above conjugation polymers increase the stability of GSNO, the S–NO bond in solution still runs the risk of breakdown. Using the prepared GSNO conjugated polymers to fabricate a nano/micro-structured formulation can be a better choice, which may further enhance the stability, sustain or control release of NO and delivery of NO to the target sites. GSNO reacts with the azlactone groups of di-block copolymers consisting of oligoethylene glycol-methacrylate and 2-vinyl-4,4-dimethyl-5-oxazolone monomer, then the polymeric nanoparticles are prepared by a self-assemble process [68]. The efficiency of NPs in delivering GSNO is improved due to their ability to stabilize NO in aqueous media. This increased stability allows for more efficient intracellular delivery, as demonstr.ated by preliminary in vitro experiments with neuroblastoma cells. In order to augment the anti-tumor activity, the co-delivery of cisplatin in combination with the NO releasing nanocarriers is performed on a type of cancer cell line called a neuroblastoma BE(2)-C The result indicates that the presence of NO increases the effectiveness of the drug.

In another study, the conjugation of GSNO (a potent agent for treating infected wounds as a hydrophilic small molecule) and PLGA (a material usually used in forming sustained delivery systems as a hydrophobic macromolecule) (Figure 8) helps to minimize the loss of GSNO during the nanoparticle preparation process (O/W emulsion-solvent evaporation method), resulting in a sufficient loading efficiency of GSNO [69]. Additionally, GSNO-PLGA NPs are able to release NO in two phases: a burst release (3 min) from the surface and a sustained release (11.27 h) from the interior of the NPs. This makes GPNPs more effective than GSNO alone in treating bacterial infections through increased interaction with bacteria.

For controlled release purpose, GSNO is conjugated on the surface of porous silicon nanoparticles (pSiNPs) as outlined in Figure 9. In this study, the release of NO is triggered by adding ascorbic acid. The released NO is able to kill both Gram-positive (Staphylococcus aureus) and Gram-negative bacteria (Escherichia coli) without harming mammalian cells. This suggests that these NO-releasing nanoparticles could be used to treat chronic wounds and other bacterial infections.

To treat multidrug resistance of cancer cells during chemotherapy, GSNO is co-delivered with doxorubicin (DOX) in polymeric nanoparticles [71]. As shown in Figure 10, GSNO is attached to the activated hydroxyl group of poly-(ethylene glycol)-block-poly (propylene sulfide) (PEG-PPS), followed by loading DOX into the functionalized amphiphilic copolymer (PEG-PPS-GSNO) nanoparticles through a double emulsion-solvent evaporation method (water in oil in water). These multifunctional nanoparticles have a high loading capacity for nitric oxide (NO), good stability and are able to release DOX when exposed to reactive oxygen species (ROS). It is also able to release NO in a sustained manner when exposed to glutathione (GSH). This NO reversed the chemo-resistance in HepG2/ADR cells, which allows for increased accumulation of DOX. The presence of GSNO enhanced the therapeutic efficiency of DOX to cancer cells. The co-delivery system of GSNO and DOX exhibits selective toxic ability to chemo-resistant cancer cells instead of healthy cells.

All in all, the conjugation of GSNO with polymer before the formulation step shows very good properties of increasing GSNO stability, reducing burst release and prolonging release. Nevertheless, the conjugation manipulation is complicated, hard to prove the successful conjugation to polymer and potentially toxic, d by reagents or residual organic solvent. Besides, the conjugation yield is limited, which means an uneconomic method of##f GSNO encapsulation.

## 5. Conclusions

In summary, the GSNO conjugated nano/micro-formulation has the advantage of extending the half-life of GSNO and making it easier to overcome the disadvantages of small hydrophilic molecules (easy to arrive at diffusion-burst release), which results in a sustained release profile of GSNO and achievement of greater efficiency. However, this strategy may have drawbacks of toxicity induced by reagents or residual organic solvent. Furthermore, to conjugate GSNO with polymers involves complicated synthesis and purification processes, which leads to higher risk of decomposition of the S-NO bond of GSNO, much more energy consumption and greater loss of GSNO, depending on the yields of the conjugation step. The strategy of *S*-nitrozation of GSH loaded nano/micro-formulation can avoid S-NO bound cleavage during the formulation preparation process, but the GSNO encapsulation efficiency is limited by the poor presence of active and available thiol groups in the particles and the diffusion situation of the nitrozation reagent. The direct GSNO loaded strategy is independent from chemical reactions, which makes the manufacturing process easier and avoids limited *S*-nitrozation yield. However, it may lead to GSNO decomposition throughout the formulation process. In conclusion, all these three strategies have merits and drawbacks. Different strategies can be chosen depending on the requirement of the application. If the formulation preparation process is mild, such as without heating manipulation or strong mechanical energy intervention, the direct GSNO encapsulation strategy will be the best choice. If sustained release of GSNO is the most important property of the delivery system, the conjugation strategy would be the best candidate. The *S*-nitrozation of the GSH loaded formulation strategy is fit for the delivery system, which may lead to the cleavage of the S-NO bond during the preparation procedure.

## Figures and Tables

**Figure 1 nanomaterials-13-00224-f001:**
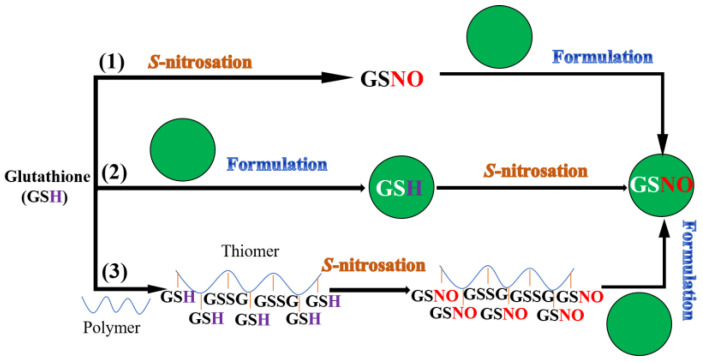
Schematic illustration of GSNO encapsulation strategies.

**Figure 2 nanomaterials-13-00224-f002:**
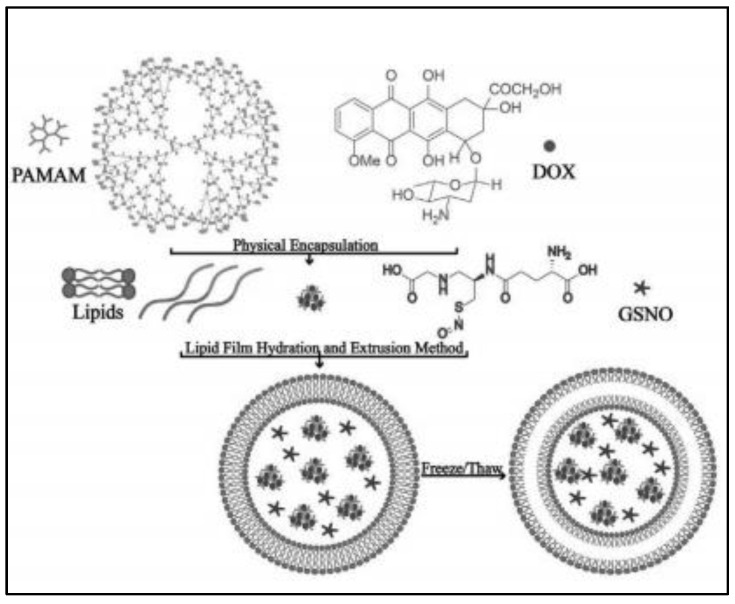
Scheme of ultrasound responsive liposome synthesis process [20].

**Figure 3 nanomaterials-13-00224-f003:**
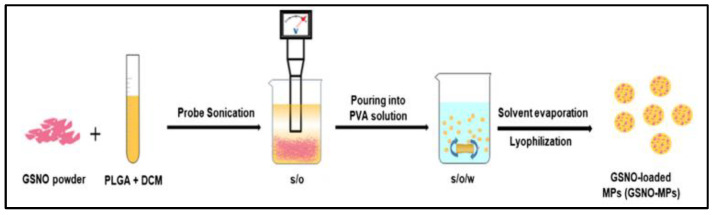
Schematic illustration of preparation of GSNO-MPs [36].

**Figure 4 nanomaterials-13-00224-f004:**
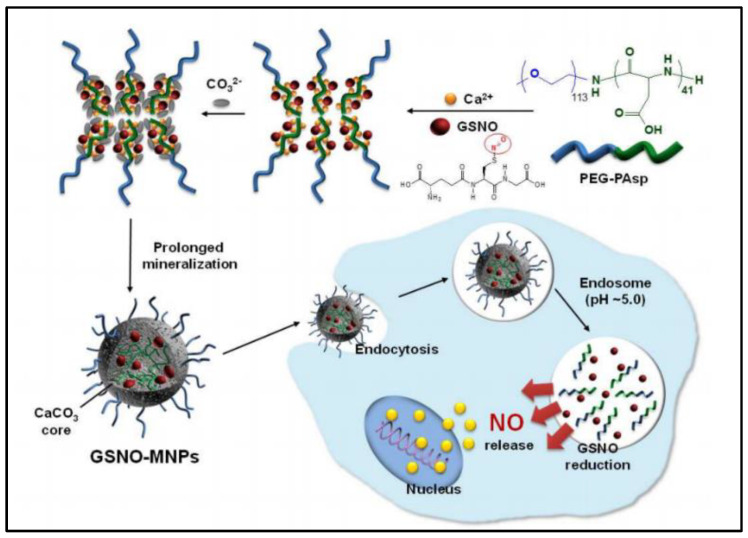
Scheme of preparation of GSNO−loaded CaCO_3_−mineralized nanoparticles [41].

**Figure 5 nanomaterials-13-00224-f005:**
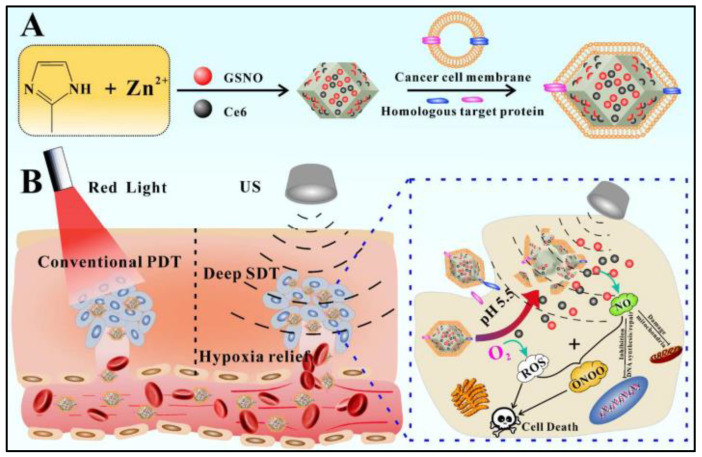
Scheme of the fabrication of the dual pH/ultrasound response biomimetic ZIF-8@Cytomembrane nanoplatform and gas-sonodynamic combined therapy. (**A**): The encapsulation strategy the formulation; (**B**): The release mechanism of the formulation [44].

**Figure 6 nanomaterials-13-00224-f006:**
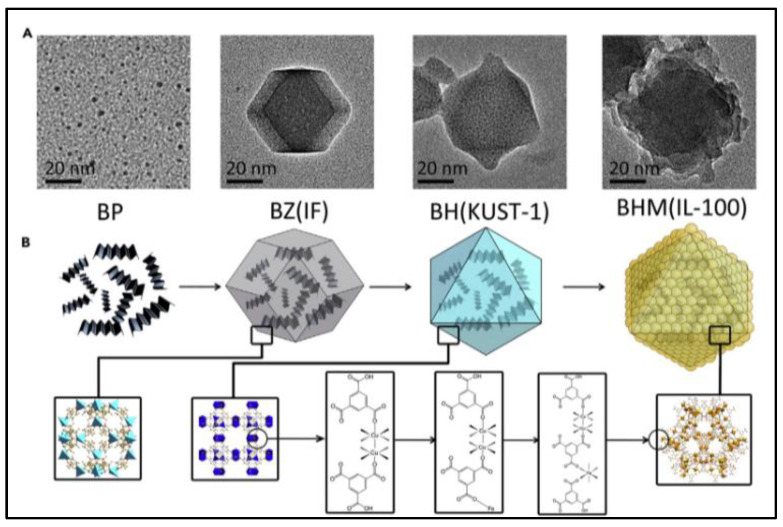
Schematic illustration of synthesis and characterization of core-shell MOF (HKUST-1@MIL-100(Fe)) nano-formulation. (**A**): Transmission electron microscope (TEM) images; (**B**): topological structures [45]. BP: black phosphorous quantum dots; BZ: black phosphorous quantum dots loaded ZIF-8 nanoparticles; BH: black phosphorous quantum dots loaded HKUST-1 nanoparticles. BHM: black phosphorous quantum dots loaded HKUST-1@MIL-100(Fe) nanocomposites.

**Figure 7 nanomaterials-13-00224-f007:**
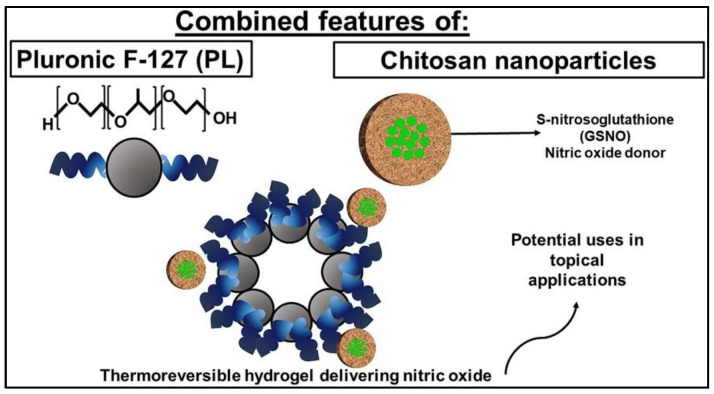
Schematic illustration of GSNO loaded chitosan nanoparticles in pluronic F-127 hydrogel [52].

**Figure 8 nanomaterials-13-00224-f008:**
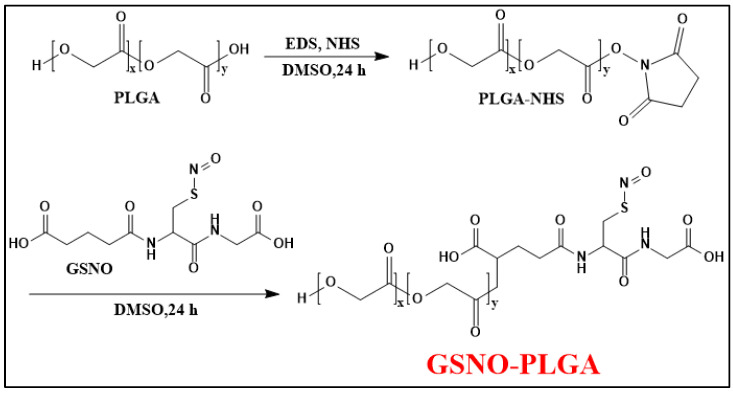
Synthesis scheme of GSNO-conjugated poly (lactic-co-glycolic acid) (GSNO-PLGA) [69].

**Figure 9 nanomaterials-13-00224-f009:**
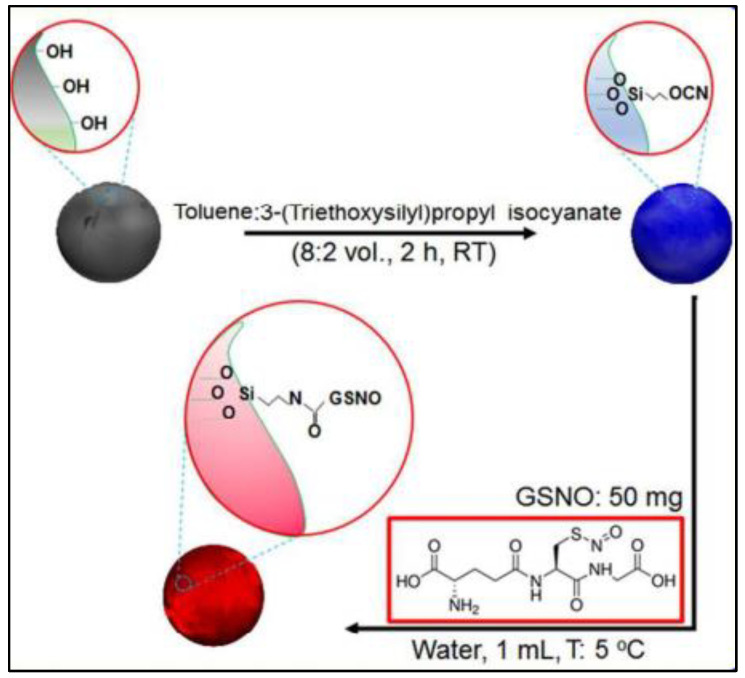
Schematic of fabrication of GSNO/pSiNPs [70].

**Figure 10 nanomaterials-13-00224-f010:**
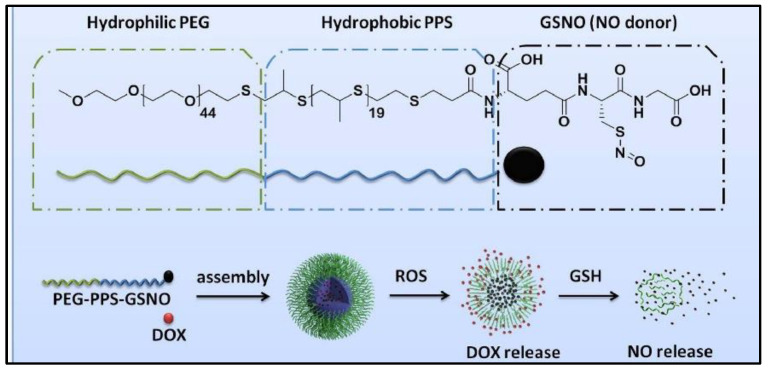
The scheme of co-assembly process of the conjugation copolymers [71].

**Table 1 nanomaterials-13-00224-t001:** Lists of direct GSNO loaded nano/micro-formulations without nanocomposites.

Formulation Type	Preparation Methods	Formulation Efficiency	Applications	Limitations	References
liposomes	solvent-spherule evaporation	Sustained release, targeted to macrophage, enhanced NO uptake and antibacterial effect	Anti-bacteria	Low encapsulation efficiency, unclear stability and toxicity of liposome	[19]
liposomes	thin film evaporation combined frozen-thaw method	Controlled release (ultrasound responsive), promoted cellular internalization and co-delivery with DOX enhanced anti-tumor effect	Anti-cancer	Unclear encapsulation efficiency	[20]
polymer nano/micro-particles	W(S)/O/W double emulsion-evaporation process	Slightly sustained release and increased intestinal permeability	/	Unclear effectiveness	[24]
polymer nanoparticles	W/O/W double emulsion-evaporation process	Sustained release and enhanced intracellular delivery of GSNO	Immune defenses	Not long enough prolonged release	[22,30]
polymer microparticles	spray-drying method	High encapsulation efficiency, controlled release (pH-sensitive)	Anti-inflammation	Non-sustained release	[31]
polymer microparticles	S/O/W emulsion-solvent evaporation method	Sustained release and enhanced antibacterial effect	Anti-bacteria	Difficult to scale up	[36]
polymer microparticles	in-situ O/W emulsion	Sustained release, reduced burst release and	Reduce arterial pressures	Uncontrollable size and potential side effects	[37,38]
CaCO_3_ nanoparticles	anionic block copolymer-templated mineralization	Increase GSNO stability, controlled release (pH-sensitive) and co-delivery with DOX enhanced anti-tumor effect	Anti-cancer	Not able to release NO quickly enough at physiological pH and unclear in vivo efficacy and safety	[41]

## Data Availability

Not applicable.

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
