# Peer review of "A Mini Review of S-Nitrosoglutathione Loaded Nano/Micro-Formulation Strategies"

_nanomaterials, 2023, doi:10.3390/nano13020224_

Round 1

Reviewer 1 Report

The authors present an interesting and comprehensive review of the encapsulation of S-nitrosoglutathione (GSNO) in different formulations of both nanometric and micrometric size. However, the authors should make several changes to make the paper suitable for publication in Nanomaterials.

Major points

They should emphasise more on the importance of doing this review and what benefits it will provide to the scientific community (they do so but in a very brief and shallow way).

Minor points

They must follow the guidelines of the journal: for example, figure caption 8 is not in the proper format.

The quality of the figures needs to be improved. For example, figures 8 and 9 are barely visible.

English language needs to be revised

Author Response

Thank you for your letter and for the reviewers’ comments concerning our manuscript for publication in Nanomaterials. These comments are valuable and very helpful for revising and improving our paper. We have studied comments carefully and have made corrections point-to-point by using “track changes” function. We hope the present manuscript meet with approval. The responds to the reviewers’ comments are as follows:

Reviewer #1:

The authors present an interesting and comprehensive review of the encapsulation of S-nitrosoglutathione (GSNO) in different formulations of both nanometric and micrometric size. However, the authors should make several changes to make the paper suitable for publication in Nanomaterials.

Major points

They should emphasis more on the importance of doing this review and what benefits it will provide to the scientific community (they do so but in a very brief and shallow way).

Author reply: We thank the reviewer for the time and effort that you have put into reviewing the previous version of the manuscript. According to the reviewers suggestion, line 48-57 was added to emphasis the importance of this review.

Minor points

They must follow the guidelines of the journal: for example, figure caption 8 is not in the proper format.

Author reply: We are very grateful for the reviewer’s suggestion. We tried to correct the format of all the figures so that it can be presented in the right way.

The quality of the figures needs to be improved. For example, figures 8 and 9 are barely visible.

Author reply: Thanks a lot for your suggestion. We tried to increase the quality of figures 8 and 9 to make them more visible. We hope that will meet the requirements of the journal.

English language needs to be revised

Author reply: Thank you very much for your suggestion. We tried to read again the whole text and did some correction by using “track changes” function.

Reviewer 2 Report

Authors need to focus on the following points:

1. In each part, only some examples were discussed in detail about the design and mechanism, but  what is the major challenge for those technologies, such as the limitation for each example or the challenge for the whole approach? What problems are limiting the development of the method?

2. In the discussion, what is the advantage and disadvantage of each method compared to the current method?

3. One table should with added on current status till date.

4. Figure 2, 3, ......and so on are just collections of images from several works. It is recommended that the author can draw high-quality diagrams to illustrate the big picture and show a few examples.

Author Response

Thank you for your letter and for the reviewers’ comments concerning our manuscript for publication in Nanomaterials. These comments are valuable and very helpful for revising and improving our paper. We have studied comments carefully and have made corrections point-to-point by using “track changes” function. We hope the present manuscript meet with approval. The responds to the reviewers’ comments are as follows:

Reviewer #2:

Authors need to focus on the following points:

  1. In each part, only some examples were discussed in detail about the design and mechanism, but what is the major challenge for those technologies, such as the limitation for each example or the challenge for the whole approach? What problems are limiting the development of the method?

Author reply: Thanks a lot for the reviewer’s useful suggestion. According to the reviewer’s suggestion, we add one paragraph in the end of each strategy to state the major challenge of each strategy. (line 314-342, 380-390 and 459-464).

  1. In the discussion, what is the advantage and disadvantage of each method compared to the current method?

Author reply: Thank you very much for your question. The current methods are what we discussed in this review which refers to 3 strategies of GSNO encapsulation. A comparison between different strategies with indication both advantages and disadvantages is made in the last paragraph. We also suggest the researchers how to choose the appropriate strategies according to their needs.

  1. One table should with added on current status till date.

Author reply: We are very grateful for the reviewer’s suggestion. According to the suggestion, a table was added after line 342.

  1. Figure 2, 3, ......and so on are just collections of images from several works. It is recommended that the author can draw high-quality diagrams to illustrate the big picture and show a few examples.

Author reply: Many thanks to the reviewer’s suggestion. The big picture is showed in figure 1 to summarize the current 3 strategies of GSNO encapsulation. The remaining figures are what we chose the representative ones.

Round 2

Reviewer 1 Report

The review has improved a lot with the changes made but:

The quality of the figures needs to be improved. For example, figures 8 and 9 are barely visible.

English language needs to be revised

Author Response

Thanks again for your letter and the reviewer’ latest comments concerning our manuscript for publication in Nanomaterials. We have studied comments carefully and have made corrections by using “track changes” function. We hope the present manuscript meet with approval. The response to the reviewer’ comments are as follows:

Reviewer #1:

The review has improved a lot with the changes made but:

The quality of the figures needs to be improved. For example, figures 8 and 9 are barely visible.

Answer: We are very grateful for your suggestion. After the discuss with the co-authors, we think that the original figure 8 is not a very representative method used in GSNO conjugation which may be not necessary to shown in this review. Thus, we decided to delete the original figure 8.

As to figure 9 (changed to figure 8 in the latest version of the manuscript), we tried to re-draw the diagram to make it more visible. We hope that will match the requirement of the journal. Thanks again for your useful advice.

English language needs to be revised

We apologize for the language problems in the manuscript. The language presentation was improved with assistance from an English fluent speaker with appropriate research background.

The revised manuscript can be found in the attachment.

Many thanks and Happy New year.